# Healthcare providers' experiences and recommendations for antiretroviral therapy adherence in Ghana: A facility-based phenomenological study

Elvis Enowbeyang Tarkang[1,2,3], Emmanuel Manu [1*], Fortres Yayra Aku[4], Joyce Komesuor[1], Veronica Charles-Unadike[1], Nelisiwe Khuzwayo[2], Judith Anaman-Torgbor [5]

1 Department of Population and Behavioural Sciences, Fred N. Binka School of Public Health, University of Health and Allied Sciences, Ho, Ghana, 2 Discipline of Public Health Medicine, School of Nursing and Public Health, University of KwaZulu-Natal, Durban, South Africa, 3 HIV/AIDS Prevention Research Network Cameroon, Kumba, Cameroon, 4 Department of Epidemiology and Biostatistics, School of Public Health, University of Health and Allied Sciences, Hohoe, Ghana, 5 Department of Public Health Nursing, School of Nursing and Midwifery, University of Health and Allied Sciences, Ho, Ghana

* emanu@uhas.edu.gh

## Abstract

### Background

In Ghana, poor treatment adherence among persons living with HIV (PLHIV) affects the effective treatment and management of HIV/AIDS. A key limitation in addressing the antiretroviral therapy (ART) adherence challenge is the skewed nature of the existing literature on the topic, as much of the research on ART adherence has focused on patient-reported experiences, failing to capture the perspectives of healthcare providers involved in ART delivery.

### Objective

We ascertained healthcare providers' perspectives on ART adherence among PLHIV and their proffered recommendations for improved ART retention in the Volta Region of Ghana.

### Methods

A total of eighteen healthcare providers offering ART services to PLHIV in the Volta Region of Ghana were purposefully recruited from five HIV sentinel sites and interviewed. The data was thematically analyzed using ATLAS.ti.

### Results

The health workers' experiences were categorized under seven thematic areas. These included difficulties in accessing healthcare providers, improvements in

**Data availability statement:** The raw data for this article will not be shared to protect the anonymity of the participants, as was agreed upon with the study participants. However, reasonable anonymous raw data can be requested from a third party within a period of five years via email: mananga@uhas.edu.gh, or from the Research Ethics Committee of the University of Health and Allied Sciences, Ho; rec@uhas.edu.gh, where the data will be kept on a dedicated desktop computer protected with a special password.

**Funding:** The author(s) received no specific funding for this work.

**Competing interests:** The authors have declared that no competing interest exist.

**Abbreviations:** ART, Antiretroviral Therapy; ARV, Antiretroviral; PLHIV, People Living with HIV; AIDS, Acquired Immunodeficiency Virus; UNAIDS, Joint United Nations Programme on HIV/AIDS; NGO, Non-Governmental Organization; GSS, Ghana Statistical Service; SDG, Sustainable Development Goal; CD4, Cluster of Differentiation; WHO, World Health Organization.

adherence to treatment, people who have interrupted treatment often being new clients, general improvements in health outcomes, financial challenges, persistent stigmatization, and varied levels of family awareness about clients' status. The reasons for treatment interruption were categorized into fifteen areas. These factors included appointment intervals, healthcare workers, medication shortages, long waiting times, clients' distance from the facility, forgetfulness, drug characteristics, financial challenges, stigmatization, drug side effects, lack of donor support, alternative treatment choices, misconceptions about HIV cures, and the location of the ART clinic within the health facility. Participants recommended making health facilities/ART sites accessible to communities, increased advocacy on HIV by PLHIV, extended appointment intervals, continuous availability of antiretroviral drugs (ARVs), community support in the treatment of HIV, improved counseling, public education on HIV to reduce stigmatization, government support, home dispensing of ARVs, regulating the activities of traditional herbalists and faith-based healers, and the encouragement of religious bodies in ART adherence education for improved ART uptake. The ART-related challenges identified by the healthcare providers should be addressed by both the Ghana Health Service and the Ghana AIDS Commission to improve ART uptake and adherence in the Volta Region and across the country if Ghana is to end the HIV pandemic by 2030, as set by UNAIDS.

## Introduction

HIV/AIDS remains a significant public health challenge in sub-Saharan Africa, with the region accounting for over two-thirds of the global HIV burden [1]. In the year 2023, 39.9 million people were living with HIV globally [2]. Of this number, 26 million people were living with the virus in Africa [3], with Ghana having 354,927 of the global burden [4].

Despite this alarming prevalence, Ghana has demonstrated a strong commitment to ending the HIV pandemic by ensuring access to antiretroviral therapy (ART) for people living with HIV [5]. The UNAIDS has set the 95-95-95 target by 2030. This means that at least 95% of people living with HIV know their HIV status, at least 95% of people who know their HIV status are on treatment, and at least 95% of people on treatment have a suppressed viral load [6]. The country has made substantial progress in scaling up ART coverage, with 68% of people living with HIV receiving treatment in 2021 [6].

However, challenges related to ART adherence persist, affecting the effectiveness of HIV treatment across the African continent, including Ghana. One of the major challenges facing ART in Africa is poor adherence to treatment regimens [7]. According to Gils [8], sub-optimal adherence to ART is a significant barrier to achieving viral suppression and improving health outcomes among people living with HIV. In some African countries, only 67% of people living with HIV have access to ART, highlighting gaps in treatment coverage [6]. Also, HIV-related stigma and discrimination remain

prevalent in many African communities, deterring individuals from seeking HIV testing, treatment, and adhering to ART regimens [9]. Stigma and discrimination have also been associated with lower rates of ART adherence across various settings [10].

In Ghana, and particularly in the Volta region, ART adherence remains a challenge hindering the effectiveness of HIV treatment and efforts to achieve viral suppression among people living with HIV (PLHIV) [11]. Tarkang et al. [12] revealed that only 51.2% of PLHIV in the Volta region had optimal adherence to ART. Poor adherence can lead to treatment failure, drug resistance, and increased morbidity and mortality among PLHIV [13]. One of the main limitations in addressing ART adherence challenges among PLHIV in Ghana and Africa is the biased nature of the existing literature on the topic [14]. For instance, much of the research on ART adherence on the continent has focused on the client perspective or patient-reported experiences of ART regimes [15,16], with few studies capturing the voices of healthcare providers. In one of such studies on healthcare providers' perspectives on ART challenges, Lahai and colleagues [17] found stigma and discrimination, frequency of medication, use of traditional medicine, lack of money for food and transport, work barriers, inadequate medicines and test kits, limited health workers, and long distances to clinics as barriers to ART adherence in Sierra Leon. In Uganda, healthcare givers reported challenges related to clients being away from home and therefore away from their pill supply, challenges picking up refills, and related to food sufficiency as factors affecting ART adherence. Healthcare providers also identified access-to-care barriers as well as other determinants such as alcohol/alcoholism, stigma, and lack of understanding about the importance of adhering to ART [18].

In Ghana, research on ART adherence has mainly captured the perspectives of PLHIV [12,13,19–26], neglecting the perspectives of healthcare providers on the subject, making the narrative incomplete. Healthcare providers' experiences and recommendations could help improve ART uptake and adherence among PLHIV in the country, as their direct interactions with PLHIV could provide valuable insights into the challenges faced by clients and the healthcare system to inform tailored interventions for ART adherence among PLHIV [27]. Furthermore, healthcare providers can offer recommendations based on their expertise and experience, proposing strategies to enhance patient education, address stigma and discrimination, improve social support systems, and optimize HIV-related healthcare delivery models [28]. Therefore, we delved into the healthcare providers' perspectives on ART adherence among PLHIV aged 18 years and above in the Volta region of Ghana, based on the framework (Fig 1) that suggests their views on clients' reasons for ART interruption and their recommended actions could guide strategies to enhance ART retention, leading to better health outcomes for PLHIV, following the qualitative reporting guidelines of O'Brien et al. [29].

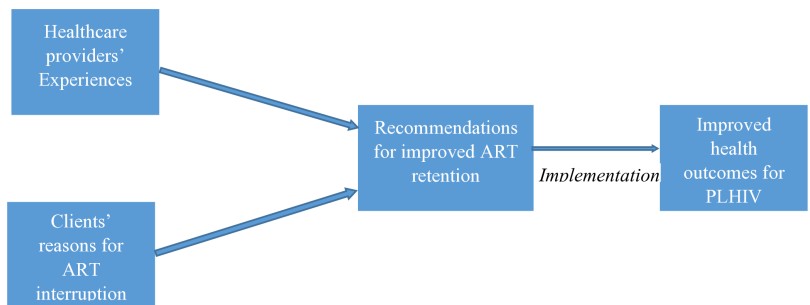

**Fig 1. Conceptual framework of the study.**

## Methods

### Context

The study took place in the five designated HIV sentinel sites that offer ART services in the Volta Region of Ghana. These sites include the Ho Municipal Hospital sentinel site, the Volta Regional Hospital sentinel site, Kpando Municipal sentinel site, the Ketu North sentinel site, and the Aflao sentinel site. The Volta Region, with Ho as its capital, is one of the sixteen regions of the country. The region has an estimated population of 1,659,040 [30] and shares borders with the Oti Region, the Gulf of Guinea, the Eastern Region, and the Republic of Togo to the north, south, west, and east respectively, located between longitudes 000 15'W and 100 15'E and latitudes 600 15'N and 800 45'N [31].

### Study design

This study was conducted from 10th March 2021–15th July 2021, using the descriptive phenomenological qualitative design to uncover the experiences of HIV/AIDS healthcare providers in the Volta Region of Ghana and their recommendations for improved ART uptake. With descriptive phenomenology, a phenomenon is described for thorough understanding instead of investigating its causal relationship [32], as the aim of the study was to understand the subjective experiences of healthcare providers regarding clients' ART uptake and adherence [33].

### Participants characteristics

The study was conducted among eighteen (18) HIV/AIDS healthcare providers in five HIV sentinel sites of the Volta Region of Ghana, with a minimum HIV/AIDS-related working experience of two (2) years. The sentinel sites were purposefully chosen as they were the sites with a combined total of more than 1000 active HIV/AIDS cases at the time of the study. Healthcare providers who had worked with HIV/AIDS clients for up to two (2) years but were not at their posts or could not be reached for interviews, and those who declined to participate in the study, were excluded. Thus, participants were purposefully recruited based on their involvement in HIV/AIDS care and their experience in managing HIV/AIDS cases at various sentinel sites.

### Researchers' characteristics and reflexivity

The research team comprised two Associate Professors experienced in qualitative research (EET & JAT), three Senior Lecturers (EM, JK & VC), and a PhD student (FYA), all with qualitative research. To ensure that their biases did not influence the interviews and to minimize subjectivity, each interviewer was accompanied by two research team members during the interviewing process, whose duty was to ensure that the interviews did not deviate significantly from their intended objectives.

### Sampling procedure

Purposive sampling, which involved HIV healthcare providers, comprising physicians, pharmacists, and nurses who offered ART and counselling services to PLHIV and met the inclusion criteria of a minimum of 2 years of HIV care provision in any of the five sentinel sites, was used to recruit participants for the study. Eighteen out of the thirty healthcare providers who were strictly involved in HIV/AIDS-related activities were interviewed, after saturation had been achieved. In each of the five sentinel sites, the list of healthcare providers that offered ART services was obtained from the administration office. Based on the availability and willingness to partake in the study during the data collection period, qualified healthcare providers per the inclusion criteria were interviewed until data saturation was reached. Among homogeneous populations such as healthcare workers, research has shown that data saturation could be reached with a sample size of twelve (12) people [34], which was the case in this study, as saturation was reached at the 18th interview, hence the sample size for the study.

## Data collection procedure

The data were collected using an in-depth qualitative interview guide. The interview guide explored the socio-demographic characteristics of the participants, their experiences as HIV/AIDS healthcare providers, their perceptions of the reasons ART clients interrupt their treatment or adhere to their treatment, and their preferred solutions for maximizing ART adherence. The guide was pre-tested among two (2) HIV/AIDS healthcare givers from a sentinel site in the Oti region that was not part of the study. This allowed for clarifying any ambiguous questions before the actual data collection. The interviews were moderated by two trained data collectors who had completed their master's degrees in public health, under the supervision of the research team. Each interview lasted an average of 45 minutes. The interviews were conducted in English as all participants had a tertiary level of education.

**Ethical considerations.** Data were collected after the researchers obtained an ethical clearance certificate from the Ethical Review Committee of the University of Health and Allied Sciences, Ho, Ghana (UHAS-REC A.2[1]20-21). The study procedures were conducted per the relevant guidelines and regulations outlined in the Declaration of Helsinki [35]. Permission to conduct the study was also obtained from the Volta Regional Directorate of the Ghana Health Service and the management of the health facilities involved in the study. Written informed consent was obtained from all the participants in this study. The confidentiality and anonymity of the participants were assured by assigning codes to participants instead of using their identities. Also, the interviews were conducted in secluded areas to avoid the glare of other colleagues and clients.

## Data analysis

Data were transcribed and compiled into Microsoft Word files from which codes and themes were developed, guided by the steps of descriptive phenomenological data analysis proposed by Colizzi [36]. The Atlas ti. v7.5 software was used to develop the codes and themes. The inductive coding process, whereby the codes were determined from the data instead of having pre-determined codes [37], was followed. To reduce subjectivity in the analysis process, the data were analyzed by two of the research team members [JAT and EM]. These steps include: (1) repeated reading of the transcripts to obtain a general sense of the content, (2) extraction and recording of significant statements that pertained to the study objectives from the transcripts, (3) formulation of meaning from the significant statements, (4) sorting of meanings into categories, clusters of themes and sub-themes, (5) integration of the findings into an exhaustive description of the phenomenon under study under various themes, and (6) finally, member checking was done to validate the descriptive results. The data analysts met after the coding process to discuss and verify the accuracy of the emerging themes and the meaning of each theme. A theme was not accepted until an agreement between the two analysts was reached.

## Rigour

Following the four elements of trustworthiness listed by Elo et al. [38] the credibility of the study findings was ensured, first, by openly communicating with HIV/AIDS healthcare providers about the nature and purpose of the study, which allowed them to be interviewed without hesitation to provide information to the research team. Additionally, peers with qualitative expertise at the Fred N. Binka School of Public Health were contacted during the design and conduct of the study for their input. Thus, we achieved dependability by incorporating the input of professional qualitative researchers on the research team and peers with qualitative expertise who evaluated our processes, transcripts, and findings to minimize bias [38]. Furthermore, a thorough and detailed description of our methods could ensure that our study could be replicated in similar settings, making our results transferable. Lastly, confirmability was maintained by ensuring that participants reviewed the transcripts and results for approval before submission for publication.

## Results

### Demographic characteristics *of* participants

According to Table 1, twelve (67%) of the participants were male, and six (33.0%) were female. A total of 67% were married, and 88.9% were either diploma or bachelor's degree holders. Most of the participants (44.4%) were nurses and 44.4% of them had less than five years of practice at their current facility and experience in ART administration.

### Thematic findings

Three broad themes emerged from the analysis. These were the experiences of healthcare providers, reasons for ART interruption, and recommendations for improved ART adherence. The experiences of the health workers were categorized under two sub-themes, positive experiences and negative experiences. The positive experiences included improvements in adherence to treatment and general improvements in health outcomes. The negative experiences include difficulties in accessing healthcare providers, people who have interrupted treatment often being new clients, financial challenges, persistent stigmatization, and varied levels of family awareness about clients' status. The reasons for treatment interruptions were categorized into fifteen areas. These factors included appointment intervals, healthcare workers' attitudes,

**Table 1. Demographic characteristics of participants.**

| Background Characteristics | Frequency (n = 18) | Percentage (%) |
|---|---|---|
| *Sex* | | |
| Female | 6 | 33.0 |
| Male | 12 | 67.0 |
| *Age* | | |
| 20-30 | 2 | 11.0 |
| 31-40 | 10 | 56.0 |
| 41-50 | 4 | 22.0 |
| 51-60 | 2 | 11.0 |
| *Marital Status* | | |
| Married | 12 | 67.0 |
| Single | 6 | 33.0 |
| *Level of Education* | | |
| Diploma/bachelor's degree | 16 | 88.9 |
| Master's degree | 2 | 11.1 |
| *Occupation* | | |
| General Medical Officer | 4 | 22.2 |
| Nurse | 8 | 44.4 |
| Pharmacist | 4 | 22.2 |
| HIV Case Manager | 2 | 11.1 |
| *Duration of Practice in Facility* | | |
| <5 Years | 8 | 44.4 |
| 5–10 Years | 4 | 22.0 |
| >10 Years | 6 | 33.5 |
| *Duration of Time Administering ARV* | | |
| <5 Years | 8 | 44.4 |
| 5–10 Years | 6 | 33.3 |
| >10 Years | 4 | 22.2 |

medication shortages, long waiting times, clients' distance from the facility, popularity, forgetfulness, drug characteristics, financial challenges, stigmatization, drug side effects, lack of donor support, alternative treatment choices, misconceptions about HIV cure, and the location of the ART clinic at the health facility. Participants suggested actions like improving accessibility of health facilities/ART sites to communities, enhancing HIV advocacy by PLHIV, extending appointment intervals, ensuring continuous ARV availability, fostering community support for HIV treatment, enhancing counseling, conducting public education on HIV to combat stigmatization, providing government support, enabling home ARV dispensing, regulating traditional herbalists and faith-based healers' activities, and involving religious organizations in ART adherence education for better ART uptake, as detailed in Table 2.

## Experiences with ARV administration

The experiences of the health workers were categorized under two sub-thematic areas: positive and negative experiences. Two positive experiences and five negative experiences were reported. The positive experiences include improvements in adherence to ART and general improvements in the health outcomes of PLHIV. The negative experiences included difficulties in accessing healthcare providers, people who have interrupted treatment often being new clients, financial challenges, persistent stigmatization, and varied levels of family awareness about clients' status.

### Positive experiences

**Improvement in adherence to ART.**  Six of the participants mentioned that there has been a marked improvement in ART adherence over the years, despite the challenges they face, compared to when the mass administration of ART was rolled out. One of the participants shared his view on this:

*"Hmm, there has been an improvement in adherence to the administration of ART now compared to when we started this mass administration. It seems people now understand the process."* [**Male, 40 years**]

**General improvement in health outcomes of PLHIV.**  Five of the participants also observed that the improvement in ART adherence over the years has led to a corresponding improvement in the health outcomes of PLHIV. One of them explained:

*"We are seeing people who are on the [ARVs] and they are now surviving for a longer time and giving birth to uninfected children while they are positive, which is one of the greatest gains. Formerly, people were transmitting it to their children, but now with the intervention, we are having 100% HIV-negative babies from positive mothers, which is a great achievement."* [**Male, 45 years**]

### Negative experiences

**Difficulty in opening to healthcare providers.**  Seven of the participants indicated that some of their clients find it difficult to open up to them in terms of their health needs because of mistrust. This made the work a bit challenging for healthcare providers. One of them explained:

*"Sometimes we find it difficult to get the correct information we need from them [ART clients] because they find it hard to open to us. Although we are their healthcare providers, they still don't trust us, but we try our best to convince them".* [**Female, 34 years**]

**People who have interrupted treatment are often new clients.**  Another experience the participants shared was that most often, ART persons who have interrupted treatment are the newly enrolled clients, as they might still be in denial or

**Table 2.** Summary of thematic findings of the study.

| Major theme | Sub-themes | Sample Quotes | |
|---|---|---|---|
| **Experience with ART administration** | **Positive experiences** | | |
| | • Improvement in adherence to treatment | • *"Hmm, there has been an improvement in adherence to the administration of ART now compared to when we started this mass administration. It seems people now understand the process."* [**Male, 40 years**] | |
| | • General improvement in health outcomes. | • *"We are seeing people who are on the ARVs, and they are now surviving for a longer time and giving birth to uninfected children while they are positive, which is one of the greatest gains. Formerly, people were transmitting it to their children, but now with the intervention, we are having 100% HIV-negative babies from positive mothers, which is a great achievement."* [**Male, 45 years**] | |
| | **Negative experiences** | | |
| | • Difficulty in opening to health-care providers | • *"Sometimes we find it difficult to get the correct information we need from them [ART clients] because they find it hard to open to us. Although we are their healthcare providers, they still don't trust us, but we try our best to convince them"*. [**Female, 34 years**] | |
| | • People who have interrupted treatment are often new clients. | • *"Yes, those who default are the new clients, but the ones who have been on the medication for a long time, you would hardly see any of them defaulting."* [**Male, 32 years**] | |
| | • Financial challenges. | • *"Concerning the clients, some of them do not have money to use as transportation to come for the medicine, and some also do not have money to renew their health insurance for the prophylaxis that has been prescribed for them."* [**Male, 55 years**] | |
| | • Persistent stigmatization. | • *"And then stigma too is coming down gradually, but it is not eliminated because there are still negative perceptions about HIV infection. Some of us [healthcare providers] sometimes look down on PLHIV, especially when they [PLHIV] are not cooperating with us. "* [**Male, 55 years**] | |
| | • Varied levels of family aware-ness about clients' status. | • *"I have been doing this work [HIV counseling] for six years now, and some of my clients are still not comfortable disclosing their status to anyone. They don't trust their own families, which makes life more difficult for them as they lack support systems. "* [**Male, 40 years**] | |
| **Reasons for ART interruption** | | **Sample quotes** | **Recommendations** |
| | • Shorter appointment intervals | • *"It can also play a role because, in case I give a one-month duration drug to somebody, and after the one month, the client does not have enough finances to come for another supply, it can be a factor."* [**Female, 27 years**] | • Prolonged appointment intervals |
| | • Attitude of healthcare workers | • *"Another problem is that they say the staff have been reporting them to the townspeople that they are HIV patients, and they are always questioned by such people. Because of that, some of them have ceased coming around for medication. I know of two people who left this facility for other facilities because of that."* [**Male, 35 years**] | • Enhance advocacy efforts to promote ART adherence. |
| | • Shortage of medication | • *"Concerning the drugs, when there is a shortage of medication in the facility, as I mentioned earlier, some of them do not originate from here, and their transportation to acquire the medication is also a problem, so they default because they are not sure whether there will be medication available for them when they come."* [**Female, 32 years**] | • Continuous availability of ARVs |
| | • Long waiting time | • *"I would say that the time taken to serve them is sometimes not the best. They endure long to be served and fear that while waiting, someone they know would come and meet them here, so they don't like to come back for appointments."* [**Female, 40 years**] | • Making health facilities/ART sites accessible |

*(Continued)*

**Table 2.** (Continued)

| Major theme | Sub-themes | Sample Quotes | |
|---|---|---|---|
| | • Location of the clients' residence from the facility | • *"Most of our clients are coming from far away, and they will tell you that money for transportation is a problem for them."* [**Female, 35 years**] | • Making health facilities/ ART sites accessible |
| | • Being popular | • *"Sometimes the person is a prominent figure in town and is taking the medicine here; surely, he or she will also know people in town, so that's why when they come on the first day, we ask them whether they know some people around. So, after taking the first treatment, s/he will not come back and prompt them to other facilities".* [**Male, 32 years**] | • Support from the community |
| | • Forgetfulness | • *"Forgetfulness, on the other hand, will be with the time they normally take it. Maybe the person is used to the 8:00 pm medication but might forget or be engaged in something and thereby forget about it. But later, most of them go for it but not at the designated time they are supposed to take it but later on."* [**Male, 40 years**] | • Efficient counseling services |
| | • The nature of drugs | • *"Some of them find it difficult to take medicines and sometimes, some complain about the size of the medicine, that it is too big. But we advised them to break it into two and take it."* [**Female, 35 years**] | • Efficient counseling services |
| | • Financial challenges | • *"Formally, they [the government] used to renew the [health] insurance renewals, but they [the government] have stopped renewing the health insurance for about 3 to 4 years. Some also complain about not having transportation fares."* [**Male, 45 years**] | • Support from governmental authorities |
| | • Stigmatization | • *"Err, some too I may say it is linked to the stigma. For example, some individuals may have come to the facility and seen a relative working there. Some hide it until they find out, then you know, "Oh, I saw a church member in the facility, fearing does not want that person to take my medications here again."* [**Male, 35 years**] | • Stigma reduction |
| | • Side effects of drugs | • *"Some will say that they started, they started having side effects from the drug, which is why they are unable to come. Some of these individuals may come back, in which case the healthcare providers can change their medication. However, others may not return."* [**Female, 34 years**] | • Efficient counseling services |
| | • The lack of donor support | • *"Initially when we started, the donors were assisting so that at one point in time it brought a lot of them on board, they empowered some of them [PLHIV] to do little petty trading into agriculture so that it can help them survive and come for their medication. But this support is lacking now and has brought about defaulting."* [**Female, 34 years**] | • Support from governmental authorities |
| | • Resort to alternative treatment options | • *"I had a client that was doing so well, then suddenly, she stopped coming [for her medication]. When I followed up, I was told they were at a prayer camp where they claimed to cure HIV. Another person also told me that she found an herbal medicine that cured someone of HIV, so she is no longer going to come for her treatment."* [**Male, 45 years**] | • Regulate the practices of traditional herbalists and faith-based healers to ensure they do not undermine ART adherence |
| | • Misconception of being cured of HIV | • *"Sometimes they [PLHIV] think that they have been cured when they no longer see symptoms after a long period on ART and their viral load has become undetectable. They will then tell people they never had HIV and were just being discriminated against."* [**Female, 40 years**] | • Promote the involvement of religious organizations in educating their communities about the importance of ART adherence |

*(Continued)*

| Major theme | Sub-themes | Sample Quotes | |
|---|---|---|---|
| | • The location of the ART clinic at the health facility | • "The location of the clinic [at the facility] is a problem. They [PLHIV] need their privacy. That is why we have our clinic day on Thursdays because the facility is less busy on that day compared to Mondays, Tuesdays, and Fridays, where we run surgery, gynecology, and other clinics." **[Male, 43 years]** | • Making health facilities/ ART sites accessible |
| **Recommendations for ART adherence** | • Making health facilities/ART sites accessible | • "The health system should be structured in a way that health facilities providing treatment [to PLHIV] are easily accessible [to ART clients]. Some of them don't have money to travel long distances for their appointments and medication, so if it is close to them, they will attend [attend their appointments]." **[Female 40 years]** | |
| | • Enhance advocacy efforts to promote ART adherence. | • "That one, you know you can't pass any law that anybody should be arrested, but one thing we can do is that if we can get any of the clients who can devote themselves to advertising that they have the illness, they are taking the medication, and they are doing well." **[Male 35 years]** | |
| | • Prolonged appointment intervals | • "Most of them want it for a longer period, like 6 months, so they can only visit the facility once or twice a year. Especially for those for whom distance is the cause of their default-ing, we make sure that when they come, we give them the drugs that will cover them for at least 3-6 months so that they will not feel they are wasting their time and money." **[Female 27 years]** | |
| | • Continuous availability of ARVs | • "Drug availability is a key factor. Making sure that clients always have drugs when they come [to the hospital] motivates them to adhere to their treatment and keep coming [for medication]." **[Female 40 years]** | |
| | • Support from the community | • "And also, community involvement is crucial. The chiefs and opinion leaders should always be talking about those who are being stigmatized in their families and create an avenue for them to report appropriately to the next channel." **[Male 33 years]** | |
| | • Efficient counseling services | • "The relationship that exists between us motivates them, and counseling plays a key thing that motivates and supporting them. When we check the drugs remaining for them to take, we congratulate them and encourage them they are almost done, so they should keep it up." **[Female 32 years]** | |
| | • Stigma reduction | • "To me, it is the stigma that is causing all those things [defaulting] so educating the public is crucial. Education that they should stop stigmatizing will also reduce discontinuation, that's what I think." **[Male 35 years]** | |
| | • Support from governmental authorities | • "If the authorities can provide some money for them to be given every clinic day if they come for the treatment, it will help them and also renew their [health] insurance for free." **[Female 40 years]** | |
| | • Dispensing ARVs at home | • "To me, if it can be allowed, even if the client allows us to bring their drugs to the house, it will be the best....like we can give the drugs to the community health nurses or us, the nurses, if they can provide us with TNT so that after the clinic day, the following day, those who were not able to come, we can go round and dispense it to them fine." **[Male 30 years]** | |
| | • Regulate the practices of traditional herbalists and faith-based healers to ensure they do not undermine ART adherence | • "Whenever the clients come to the facility, we have a way that we get to determine if they have visited any herbalists or not. We advise them and inform them that they might die if they continue to visit herbalists and prophets without adhering to their treatment plan." **[Male 40 years]** | |
| | • Promote the involvement of religious organizations in educating their communities about the importance of ART adherence | • "I'm a man of God and I have to say this errr… if men of God would speak to their con-gregation and let them know that medicine also plays a crucial role and was created by God, they [PLHIV] will adhere. People look up to them, and when they deteriorate to a critical state, they allow them to come and pass away in healthcare facilities, portraying the health system as if it is ungodly for a child of God to have this infection." **[Male 40 years]** | |

*(Continued)*

fear of being identified and stigmatized through their continued visits to the clinic. This was alluded to by five participants, as explained by one of them:

*"Yes, those who default are the new clients, but the ones who have been on the medication for a long time, you would hardly see any of them defaulting."* **[Male, 32 years]**

**Financial challenges.** Another experience that the participants shared with us was the continuous financial struggle PLWHIV continue to face, despite interventions to improve their economic fortunes. This made it difficult for PLHIV to adhere to treatment, as alluded to by seven participants. A male participant said:

*"Concerning the clients, some of them do not have money to use as transportation to come for the medicine, and some also do not have money to renew their health insurance for the prophylaxis that has been prescribed for them."* **[Male, 55 years]**

**Persistent stigmatization.** Ten participants noted that while there has been a decrease in the stigmatization of PLHIV, the issue still exists. Even some healthcare providers often stigmatize and discriminate against PLHIV in various health facilities, as captured in the excerpt below:

*"And then stigma too is coming down gradually, but it is not eliminated because there are still negative perceptions about HIV infection. Some of us [healthcare providers] sometimes look down on PLHIV, especially when they [PLHIV] are not cooperating with us. "* **[Male, 55 years]**

**Varied levels *of* family awareness *about* clients' status.** Another experience shared by participants was the varying levels of status disclosure among PLHIV to their immediate family members. Six participants noted that while some clients were comfortable disclosing their status to their families, some were hesitant despite encouragement. One of them explained:

*"I have been doing this work [HIV counseling] for six years now, and some of my clients are still not comfortable disclosing their status to anyone. They don't trust their own families, which makes life more difficult for them as they lack support systems. "* **[Male, 40 years]**

### Reasons for ART interruption

The reasons for treatment interruption were categorized into fifteen areas. These factors included appointment intervals, healthcare workers, medication shortages, long waiting times, clients' distance from the facility, forgetfulness, drug characteristics, financial challenges, stigmatization, drug side effects, lack of donor support, alternative treatment choices, misconceptions about HIV cures, and the location of the ART clinic within the health facility.

**Shorter appointment intervals.** Some of the participants (12) highlighted the intervals between appointments as one of the reasons for patients interrupting their ART treatment. This stems from the fact that sometimes, these intervals are short, posing a financial challenge for clients to adhere to appointments. A female participant explained:

*"It can also play a role because, in case I give a one-month duration drug to somebody, and after the one month, the client does not have enough finances to come for another supply, it can be a factor."* **[Female, 27 years]**

**Attitude of some healthcare providers.** Eight of the participants reported that the attitude of their colleague health workers often leads to patients interrupting treatment. This is because of breaching confidentiality, as some care providers gossip about their clients with colleagues or the public. One of them explained:

*"Another problem is that they say the staff have been reporting them to the townspeople that they are HIV patients, and they are always questioned by such people. Because of that, some of them have ceased coming around for medication. I know of two people who left this facility for other facilities because of that."* [**Male, 35 years**]

**Periodic shortage of drugs.** The periodic shortage of medication (ARVs) meant for the treatment of HIV was another reason that four of the participants cited non-adherence to treatment among their clients. They explained that the uncertainty surrounding drug availability during medical appointments made some clients hesitant to adhere to their appointment schedules. The quote below summarizes their views:

*"Concerning the drugs, when there is a shortage of medication in the facility, as I mentioned earlier, some of them do not originate from here, and their transportation to acquire the medication is also a problem, so they default because they are not sure whether there will be medication available for them when they come."* [**Female, 32 years**]

**Long waiting time.** Some participants (6) explained that PLHIV often interrupted their treatment because of the time they spent at the facility to be attended to. The delays in serving them caused some patients not to return for their subsequent appointments. One of them narrated:

*"I would say that the time taken to serve them is sometimes not the best. They endure long to be served and fear that while waiting, someone they know would come and meet them here, so they don't like to come back for appointments."* [**Female, 40 years**]

**The proximity of the client's residence to the facility.** The distance between clients' residences to the facility was also highlighted by five participants as a reason for patients' interruption in treatment. This is because patients may be coming from distant locations, which has economic implications for them. One of them explained:

*"For most of our clients, they are coming from far away, and they will tell you that money for transportation is a problem for them."* [**Female, 35 years**]

**Being popular.** One participant noted that PLHIV being well-known in their communities is another factor that leads to ART interruption. He explained that due to their popularity, they felt embarrassed to visit the health facility for treatment, as they could easily be recognized by someone they knew. He narrated:

*"Sometimes the person is a prominent figure in town and is taking the medicine here; surely, he or she will also know people in town, so that's why when they come on the first day, we ask them whether they know some people around. So, after taking the first treatment, s/he will not come back and prompt them to other facilities".* [**Male, 32 years**]

**Forgetfulness.** Some participants (eleven) mentioned forgetfulness as a common reason why some clients interrupt ART treatment. This group of clients either forgets their medication appointment dates or the time they are supposed to take their medication. The excerpt captures their perspectives.

*"Forgetfulness, on the other hand, will be with the time they normally take it. Maybe the person is used to the 8:00 pm medication but might forget or be engaged in something and thereby forget about it. But later, most of them go for it, but not at the designated time they are supposed to take it, but later on".* [**Male, 40 years**]

**The nature of the drugs.** Additionally, some participants (3) noted that some PLHIV interrupt their treatment because they find the ARVs too large to swallow and intentionally avoid taking them. One of the participants narrated:

*"Some of them find it difficult to take medicines and sometimes, some complain about the size of the medicine, that it is too big. But we advised them to break it into two and take it".* [**Female, 35 years**]

**Financial challenges.** Thirteen participants revealed that PLHIV mainly interrupt treatment due to financial challenges. More specifically, the lack of money to afford health insurance renewal and fund transportation to the health facility. One of them explained:

*"Formally, they [the government] used to renew the [health] insurance renewals, but they [the government] have stopped renewing the health insurance for about 3 to 4 years. Some also complain about not having transportation fares."* [**Male, 45 years**]

**Stigmatization.** Although HIV has been in existence for a long time, those living with the virus are still stigmatized. Ten of the participants mentioned that the stigma associated with HIV, leading to PLHIV, interrupted medical appointments and consequently their treatment when encountering familiar faces at the health facility. One explained:

*"Err, some too, I may say it is linked to the stigma. For example, some individuals may have come to the facility and seen a relative working there. Some hide it until they find out, then you know, "Oh, I saw a church member in the facility, fearing does not want that person to take my medications here again."* [**Male, 35 years**]

**Side effects of medications.** The side effects that arise from ART were also cited as a reason for some patients' interruption of treatment. Seven of the participants mentioned this, as narrated by one of them below:

*"Some will say that they started, they started having side effects from the drug, which is why they are unable to come. Some of these individuals may come back, in which case, the healthcare providers can change their medication. However, others may not return."* [**Female, 34 years**]

**Lack of support from donors.** One of the participants cited the lack of support from the donors as a reason why some PLHIV interrupted treatment. He explained that in the past, these donors were given a lot of financial support to PLHIV, enabling them to attend their scheduled medical appointments. However, this is currently not the case. She explained:

*"Initially, when we started, the donors were assisting so that at one point in time it brought a lot of them on board; they empowered some of them [PLHIV] to do little petty trading into agriculture so that it could help them survive and come for their medication. But this support is lacking now and has brought about defaulting."* [**Female, 34 years**]

**Some PLHIV resort to alternative treatment.** It was also mentioned (six participants) that some PLHIV think that they can be cured of the disease if they resort to alternative medicine or faith-based healing, and thus interrupt their treatment. One participant explained:

*"I had a client who was doing so well, then suddenly, she stopped coming [for her medication]. When I followed up, I was told they were at a prayer camp where they claimed to cure HIV. Another person also told me that she found a herbal medicine that cured someone of HIV, so she is no longer going to come for her treatment."* [**Male, 45 years**]

**Misconception of being cured of HIV.** Four participants mentioned that one reason for ART interruption is the misconception that PLHIV is cured when their viral load becomes undetectable. One participant said:

*"Sometimes they [PLHIV] think that they have been cured when they no longer see symptoms after a long period on ART and their viral load has become undetectable. They will then tell people they never had HIV and were just being discriminated against."* [**Female, 40 years**]

**Location of the ART unit within the health facility.** Another finding was that the location of the ART unit in the health facility is a reason why some patients interrupted their treatment. Three participants alluded to this, as explained by one of them:

*"The location of the clinic [at the facility] is a problem. They [PLHIV] need their privacy. That is why we have our clinic day on Thursdays because the facility is less busy on that day compared to Mondays, Tuesdays, and Fridays, where we run surgery, gynecology, and other clinics."* [**Male, 43 years**]

**Recommendations to prevent treatment interruption**

Participants were asked how adherence to ART could be improved in the Volta region. Eleven sub-themes emerged under this theme. These include making health facilities/ART sites accessible to communities, increased advocacy on HIV by PLHIV, extended appointment intervals, continuous availability of ARVs, community support in the treatment of HIV, improved counseling, public education on HIV to reduce stigmatization, government support, home dispensing of ARVs, regulating the activities of traditional herbalists and faith-based healers, and the encouragement of religious bodies in ART adherence education.

**Accessibility of health facilities.** Seven of the participants were of the view that making health facilities accessible to PLHIV could help improve adherence to ART. This is because PLHIV would not have to travel longer distances to access ART services. One of them narrated:

*"The health system should be structured in a way that health facilities providing treatment [to PLHIV] are easily accessible [to ART clients]. Some of them don't have money to travel long distances for their appointments and medication, so if it is close to them, they will attend [attend their appointments]."* [**Female 40 years**]

**HIV Advocacy by PLHIV.** One of the participants mentioned that improved advocacy by PLHIV is a useful tool to prevent patients from treatment interruption. He believed that if volunteer PLHIV were bold enough to advocate treatment adherence, the situation could improve.

*"That one, you know you can't pass any law that anybody should be arrested, but one thing we can do is that if we can get any of the clients who can devote themselves to advertising that they have the illness, they are taking the medication, and they are doing well."* [**Male 35 years**]

**Increasing appointment intervals.** Increasing appointment intervals were also cited by four participants as an important recommendation to prevent ART interruption. They believed that longer appointment intervals led to ART interruption, as some clients struggled financially to attend such appointments. One of them explained:

*"Most of them want it for a longer period, like 6 months, so they can only visit the facility once or twice a year. Especially for those for whom distance is the cause of their defaulting, we make sure that when they come, we give*

*them the drugs that will cover them for at least 3-6 months so that they will not feel they are wasting their time and money."* **[Female 27 years]**

**Continuous availability of ARVs.** Three of the participants recommended the continuous availability of drugs to prevent treatment interruption. They said that once clients get their medication whenever they visit the health facility, they are encouraged to adhere to treatment. The quote below summarizes their opinions.

*"Drug availability is a key factor. Making sure that clients always have drugs when they come [to the hospital] motivates them to adhere to their treatment and also keep coming [for medication]."* **[Female 40 years]**

**Community Involvement.** Community involvement was also suggested to prevent patients from discontinuing treatment. According to two of the participants, the involvement of key community stakeholders in HIV education could reduce stigma and enhance treatment adherence. One of them explained:

*"And also, community involvement is crucial. The chiefs and opinion leaders should always be talking about those who are being stigmatized in their families and create an avenue for them to report appropriately to the next channel."* **[Male 33 years]**

**Efficient counseling services.** Eleven participants recommended counseling to prevent patients from interrupting their treatment. They believed intensive counseling could encourage PLHIV to comply with treatment.

*"The relationship that exists between us motivates them, and counseling plays a key role that motivating and supporting them. When we check the drugs remaining for them to take, we congratulate them and encourage them that they are almost done, so they should keep it up."* **[Female 32 years]**

**Public education to minimize stigmatization.** Three participants suggested educating the public as well as passing laws to deter people from stigmatizing individuals with HIV. This approach, they believed, could reduce HIV-related stigma and enhance adherence to ART.

*"To me, it is the stigma that is causing all those things [defaulting] so educating the public is crucial. Education that they should stop stigmatizing will also reduce discontinuation, that's what I think."* **[Male 35 years]**

**Governmental support.** A few of the participants (two) interviewed recommended governmental support in terms of transportation, renewing health insurance, and economic empowerment for PLHIV as a measure to prevent ART interruption. One of them explained:

*"If the authorities can provide some money for them to be given every day if they come for the treatment, it will help them and also renew their [health] insurance for free."* **[Female 40 years]**

**Dispensing ARVs at home.** Home dispensing of ARVs was recommended to prevent patients from interrupting. Six participants believed that, due to stigmatization, if healthcare providers could be financially supported to deliver ARVs to their clients' homes of their clients, they would adhere to treatment. One of them narrated:

*"To me, if it can be allowed, even if the client allows us to bring their drugs to the house, it will be the best....like we can give the drugs to the community health nurses or us, the nurses, if they can provide us with TNT so that after the*

*clinic day, the following day, those who were not able to come, we can go round and dispense it to them fine.”* **[Male 30 years]**

**Regulation of traditional herbalists and faith-based healers.** One participant recommended that regulating the activities of traditional herbalists and faith-based healers would go a long way in preventing PLHIV from interrupting their treatment. He narrated:

*“Whenever the clients come to the facility, we have a way that we get to determine if they have visited any herbalists or not. We advise them and inform them that they might die if they continue to visit herbalists and prophets without adhering to their treatment plan.”* **[Male 40 years]**

**Encouragement of Medication Adherence during Religious Gatherings.** Another participant expressed the opinion that if religious leaders are encouraged to promote ART adherence during religious gatherings, such as church services, it could help PLHIV believe in the effectiveness of ART instead of believing in miracle healing. He elaborated:

*“I’m a man of God, and I have to say this err… if men of God would speak to their congregation and let them know that medicine also plays a crucial role and was created by God, they [PLHIV] will adhere. People look up to them, and when they deteriorate to a critical state, they allow them to come and pass away in healthcare facilities, portraying the health system as if it is ungodly for a child of God to have this infection.”* **[Male 40 years]**

## Discussion

In this study, we explored healthcare providers’ experiences, challenges, and recommendations for improved antiretroviral therapy adherence in Ghana. Concerning ART-related experiences, both positive and negative experiences were reported. The positive experiences include improvements in ART adherence and health outcomes among PLHIV, while the negative experiences were difficulties in accessing healthcare providers, people who have interrupted treatment often being new clients, financial challenges, persistent stigmatization, and varied levels of family awareness about clients’ status.

Improvements in ART adherence with associated improvement in overall health outcomes among clients were reported as positive ART-related experiences by HIV/AIDS care providers. Studies have shown that good provider-patient relationships, regular follow-ups, and positive interactions lead to less interruption in treatment [24,39,40]. This improvement in adherence and health outcomes results in fewer hospitalizations, reduced complications, and improved overall health [7,41–43]. Hence, training healthcare workers in communication and patient-centered care is crucial for enhancing ART adherence among PLHIV.

Concerning ART-related negative experiences, HIV/AIDS care providers reported that clients find it difficult to open up to them. Studies conducted in South Africa and the United States have shown that many HIV-positive individuals feel stigmatized by healthcare providers, leading to the concealment of their status or avoidance of care, respectively [44,45]. This can result in suboptimal treatment outcomes and an increased risk of drug resistance [46]. There is a need to build trusted relationships between healthcare providers and their clients living with HIV to improve communication between them [47].

Participants noted that those newly enrolled in ART programs are more likely to interrupt their treatment. Several studies revealed that fear of stigma and discrimination is a major barrier to ART adherence, particularly among newly diagnosed individuals [48,49]. Focusing on supporting newly diagnosed clients is thus essential for maintaining high levels of ART adherence and retention in care, which is crucial for achieving the UNAIDS 95-95-95 targets and ending the HIV/AIDS epidemic [6,48,50].

Furthermore, financial challenges on the part of PLHIV were reported by healthcare providers as an ART-related experience they have had. Effective HIV/AIDS care involves not only medication management but also addressing the

psychological, social, and economic aspects of a person living with HIV/AIDS [51]. Research has shown that the high costs associated with ART procedures place a significant financial burden on healthcare providers and patients alike [52,53], serving as a significant barrier to the access and utilization of ART services in Ghana [13]. This can potentially lead to non-adherence to appointments, treatment, and relapses [54,55]. Support from the government or NGOs for vulnerable groups could help improve treatment adherence. Also, as ART is a lifelong intervention, empowering PLHIV to be involved in income generation ventures could help improve their ART retention rate.

While there has been a reduction in the stigmatization of people living with HIV (PLHIV), the problem persists, even among healthcare providers within the health facilities. Studies have shown that healthcare settings continue to be environments where PLHIV face stigmatization and discrimination [56–58]. This can discourage PLHIV from seeking necessary healthcare services, leading to delays in diagnosis, treatment initiation, and poor adherence to ART [50]. Developing and enforcing institutional policies against stigma and discrimination in healthcare settings is recommended [59].

Healthcare providers observed diverse patterns of HIV status disclosure among PLHIV to their immediate family members. Similar studies indicate that PLHIV exhibited diverse patterns of disclosure, ranging from complete disclosure to all family members to selective or non-disclosure, depending on individual circumstances and perceived risks, which is consistent with the findings of the study [60–62]. The stigma associated with HIV/AIDS makes it difficult for many clients to disclose their HIV status to family members [63].

Reasons for ART interruption that were alluded to by participants include appointment intervals and location of clients' residences from the facility, attitudes of healthcare workers, shortage of medication, long waiting times, being busy, forgetfulness, the nature of the drugs, side effects of the drugs, the lack of donor support, resort to alternative treatment options, misconception of being cured of HIV, and the location of the ART clinic at the health facility.

The intervals between appointments and the location of clients' residences from the facility were reported by participants as the reasons for patients interrupting ART. Short intervals between appointments pose financial challenges for clients, especially those traveling long distances [64,65]. Also, long distances to ART facilities significantly contribute to treatment interruption among people living with HIV [66]. This can lead to missed appointments and treatment interruptions [39]. Establishing more ART clinics in communities could help reduce travel distances and associated costs [7]. With regards to this, discussants cited lengthening appointment intervals as an important recommendation to prevent interrupting ART. Studies have found that longer appointment intervals (3–6 months) are associated with improved ART adherence compared to shorter intervals [65,67,68]. This approach can alleviate the logistical and financial burden of frequent clinic visits [69]. Also, participants cited that making health facilities/ART sites accessible to communities will help prevent PLHIV from interrupting treatment. Different studies have reported that proximity and easy access to ART services were facilitators of adherence and a measure to prevent PLHIV from interrupting their treatment [70,71]. This may reduce the time, travel costs, and logistical barriers that PLHIV faces in accessing treatment [70,72]. Implementing decentralized ART delivery models, such as mobile clinics, to improve accessibility, could be considered by the Ghana Health Service and the Ghana AIDS Commission.

Also, it was found that the attitude of healthcare workers was a reason for ART interruption among HIV/AIDS clients. Health workers being harsh or disrespectful can erode trust and deter PLHIV from seeking care [73,74]. In Ghana, the narrative has consistently been that health workers are harsh and frequently treat patients without respect by ordering them around, yelling at them, or accusing them of lying [47]. Implementing clear institutional policies that promote respectful, patient-centered care is recommended to enhance healthcare delivery and patient outcomes. Studies indicate that healthcare facilities with well-defined patient-centred policies demonstrate improved patient satisfaction and treatment adherence rates [59,75]. These policies will create standardized frameworks guiding healthcare providers in delivering dignified care while maintaining professional standards [76,77]. Moreover, such policies ensure consistent quality care delivery regardless of provider differences or institutional constraints, leading to better health outcomes for PLHIV and enhanced experiences.

Another reason for ART interruption was the periodic shortages of medication (ARVs) meant for the treatment of HIV. Studies revealed that periodic shortages of antiretroviral drugs (ARVs) lead to treatment interruptions, drug resistance, and disease progression [8,61,78,79]. These shortages can also hinder efforts to achieve global targets like the 95-95-95 goals set by the United Nations. Moreover, Stock-outs have been identified as a significant barrier to adherence [16,80,81]. Hence, the Ghana Health Service, in conjunction with the Ghana AIDS Commission, should prioritize strengthening pharmaceutical supply chains and logistics management systems to ensure the continuous availability of ARV drugs at all levels of the healthcare system to prevent medication shortages.

Moreover, long waiting times were another reason reported by participants that led to ART interruption in the Volta Region. Several studies have revealed that extended waiting periods at clinics discourage PLHIV from accessing ART services [13,82]. Similarly, research has found that long waiting times discourage PLHIV from accessing ART services [83], which is often attributed to insufficient healthcare personnel attending to ART-related services [65]. Addressing staffing needs at ART clinics can help reduce waiting times and improve service efficiency [65]. Studies have demonstrated that adequate staffing levels significantly improve patient flow and service delivery at ART facilities. This improvement in operational efficiency not only reduces the time clients spend waiting for services but also enhances the quality of care provided. Furthermore, proper staffing ensures that healthcare workers can dedicate appropriate time to each client, leading to better treatment outcomes and increased patient satisfaction.

Furthermore, participants stated that PLHIV being prominent in their communities was another factor that led to ART interruption. Similar reports were made in several studies indicating that PLHIV who are well-known in their communities often interrupt ART due to fear of status disclosure and potential discrimination [84,85]. Concerns include social exclusion, job loss, and strained relationships [10]. To address this, strengthening confidentiality protocols in ART clinics is crucial to protect PLHIV's privacy and anonymity, particularly for well-known community members. In addition to that, home dispensing of ARVs to PLHIV was recommended as a preventive measure to improve adherence to ART among PLHIV, particularly for prominent and popular people. Several studies indicate that implementing home-based ART delivery programs can improve treatment adherence by removing the burden of frequent visits and protecting PLHIV's privacy and anonymity, particularly for well-known community members [86–88]. Therefore, the Ghana Health Service and the Ghana AIDS Commission could consider implementing and scaling up home-based ART delivery programs in Ghana, leveraging the skills and resources of community health workers and community-based health organizations.

It was also found that some clients forget their appointment dates for medications or forget the time they are supposed to take their medication, leading to ART interruption. Studies indicate that forgetfulness is a common barrier to ART adherence [89,90]. Forgetfulness can stem from various factors such as busy schedules, lack of reminders, or cognitive challenges [91]. This could lead to sub-optimal treatment outcomes, including increased viral load, disease progression, and the development of drug resistance [92]. Implementing robust reminder systems, such as alarms, SMS alerts, or pill organizers, can help clients remember their medication schedules. Studies have shown that these reminder mechanisms significantly help clients maintain their medication schedules and reduce instances of missed doses. The integration of such reminder systems provides practical support for clients in managing their treatment regimens, particularly beneficial for those with complex medication schedules or those who may struggle with consistent medication timing.

The findings of the study also revealed that some PLHIV interrupt their treatment because they perceive the ARVs as being too big to swallow. Studies have shown that the nature of the ART medication influences PLHIV to interrupt treatment [14,93]. Non-adherence due to perceptions about pill size can result in drug resistance and increased HIV transmission risk [94]. Hence, developing smaller, more easily swallowable ARV formulations could help address pill size perceptions, leading to improved ART adherence [14]. Studies have shown that pill size perception significantly influences treatment adherence, with larger pills often presenting challenges for patients [95,96]. The introduction of more easily swallowable ARV formulations could potentially improve patient compliance with medication regimens, ultimately leading to better treatment outcomes and reduced instances of therapy interruption among individuals on ART [97].

The side effects that arise from ART were also cited as a reason for some patients or PLHIV interrupting their treatment. Adverse reactions to ART medications contribute to non-adherence among PLHIV [98–100]. Enhancing patient education and counseling on the potential side effects of ART could help manage this reason for ART interruption among PLHIV [75]. Studies demonstrate that comprehensive education about potential side effects, their management, and the importance of treatment continuation despite mild symptoms helps patients better cope with their treatment journey [99,100]. Moreover, effective counseling empowers PLHIV with knowledge and strategies to manage side effects, leading to improved treatment adherence and better health outcomes [101].

Again, it was found that the lack of donor support was another reason for ART interruption among PLHIV in the current study. Discontinuation of donor funding leads to financial constraints and difficulties in accessing treatment [102–104]. As a result, PLHIV with fewer resources and who are underprivileged may decide not to attend clinical appointments. This might hurt treatment compliance and lead to relapse [55]. Advocating for increased and sustained donor funding for HIV/AIDS programs is crucial. Participants mentioned governmental support as a measure to prevent treatment interruption among PLHIV. As already alluded to in this study, transportation costs and access to healthcare have been significant barriers to adherence to ART [105]. This has been reported extensively in the literature [104,106]. Providing financial support for transportation, health insurance renewal, and economic empowerment for PLHIV can alleviate financial burdens and enable better access to HIV care and treatment [89,64].

PLHIV reportedly resorted to alternative treatment options, which were cited by discussants as a reason for ART interruption. Studies indicate that some PLHIV resort to traditional or faith-based healing, believing they can be cured without ART [51,105,107]. The belief that alternative or faith-based treatments can cure HIV may be influenced by a lack of understanding about the chronic nature of HIV or dissatisfaction with conventional medical care [105]. Sustained education on HIV treatment, particularly for newly diagnosed individuals, is important. The regulation of the activities of traditional herbalists, faith-based healers, and religious leaders by the government could prevent PLHIV from interrupting their ART regimens. The use of traditional and complementary medicine for HIV treatment has led to sub-optimal adherence to ART [108,109]. Hence, religious organizations can provide psychosocial support, counseling, and stigma reduction efforts if they are empowered and resourced [110].

Another reason for ART interruption was the misconception among PLHIV that they had been cured of the disease when their viral load reached undetectable levels. Several studies indicate that some patients stop treatment when their viral load becomes undetectable [82,83]. This belief may be reinforced by incomplete or inaccurate information from healthcare providers, community members, or other sources [86]. It can lead to the re-emergence of detectable viral loads, increased risk of disease progression, and the potential development of drug-resistant strains of HIV [111]. In addition, providing public education on HIV to reduce misconceptions about PLHIV would help clients adhere to treatment. Raising awareness about HIV/AIDS, prevention methods, and available treatment options can empower PLHIV [112]. Hence, the Volta Regional Health Directorate could embark on community-based HIV/AIDS education to create more awareness of the condition and thus minimize any misconceptions about HIV and PLHIV, leading to treatment adherence and improved health outcomes.

## Strengths and limitations of the study

The study was conducted among various cadres of healthcare workers directly involved in ART services delivery in the Volta region, making the study findings credible enough to inform policy on ART services delivery in the region. However, the main limitation of the study findings is that just as is the case with qualitative inquiry, our findings could be influenced by contextual factors within the setting where the study was conducted, hence, the findings should be interpreted and generalised with caution. Nevertheless, the minimal subjectivity maintained in the study's execution and the detailed reporting of our processes support the trustworthiness of the results.

## Conclusion

This study concludes that while healthcare providers were enthused about the improved adherence to ART and its associated improved health outcomes among clients, challenges such as poor attitude of healthcare workers, periodic shortage of drugs, long waiting periods at ART centres and long-distance travel to ART sites hamper adherence to ART in the Volta region of Ghana. Hence, both the Ghana Health Service and the Ghana AIDS Commission should strive to address these systemic challenges for improved ART uptake and adherence in the Volta Region and the country at large if the country is to end the HIV pandemic by the year 2030 as set by the UNAIDS.

## Acknowledgments

We would like to thank the operational managers at the various ART centers that were involved in this study. We are very thankful to all the participants who contributed to this study.

## Author contributions

**Conceptualization:** Elvis Enowbeyang Tarkang, Emmanuel Manu, Nelisiwe Khuzwayo, Judith Anaman-Torgbor.

**Data curation:** Fortres Yayra Aku, Veronica Charles-Unadike.

**Formal analysis:** Joyce Komesuor.

**Funding acquisition:** Judith Anaman-Torgbor.

**Investigation:** Fortres Yayra Aku, Joyce Komesuor, Veronica Charles-Unadike.

**Methodology:** Emmanuel Manu, Joyce Komesuor, Veronica Charles-Unadike.

**Project administration:** Elvis Enowbeyang Tarkang.

**Resources:** Elvis Enowbeyang Tarkang, Fortres Yayra Aku, Veronica Charles-Unadike.

**Supervision:** Elvis Enowbeyang Tarkang, Nelisiwe Khuzwayo, Judith Anaman-Torgbor.

**Validation:** Fortres Yayra Aku, Veronica Charles-Unadike.

**Visualization:** Nelisiwe Khuzwayo.

**Writing – original draft:** Emmanuel Manu.

**Writing – review & editing:** Elvis Enowbeyang Tarkang, Nelisiwe Khuzwayo, Judith Anaman-Torgbor.

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
