## [Decision Letter · Decision Letter 0]

2 Dec 2024

PONE-D-24-41127Healthcare providers' experiences and recommendations for antiretroviral therapy adherence in Ghana: a facility-based phenomenological studyPLOS ONE

Dear Dr. Manu,

Thank you for submitting your manuscript to PLOS ONE. After careful consideration, we feel that it has merit but does not fully meet PLOS ONE’s publication criteria as it currently stands. Therefore, we invite you to submit a revised version of the manuscript that addresses the points raised during the review process.

We look forward to receiving your revised manuscript.

Kind regards,

Joel Msafiri Francis, MD, MS, PhD

Academic Editor

PLOS ONE

Journal Requirements:

2. We note that your Data Availability Statement is currently as follows: 

“All relevant data are within the manuscript and its Supporting Information files.”

4. Please ensure that you refer to Figure 2 in your text as, if accepted, production will need this reference to link the reader to the figure.

5. We note that Figure 2 in your submission contain map images which may be copyrighted. All PLOS content is published under the Creative Commons Attribution License (CC BY 4.0), which means that the manuscript, images, and Supporting Information files will be freely available online, and any third party is permitted to access, download, copy, distribute, and use these materials in any way, even commercially, with proper attribution. For these reasons, we cannot publish previously copyrighted maps or satellite images created using proprietary data, such as Google software (Google Maps, Street View, and Earth). For more information, see our copyright guidelines: http://journals.plos.org/plosone/s/licenses-and-copyright.

1) You may seek permission from the original copyright holder of Figure 2 to publish the content specifically under the CC BY 4.0 license.  

2) If you are unable to obtain permission from the original copyright holder to publish these figures under the CC BY 4.0 license or if the copyright holder’s requirements are incompatible with the CC BY 4.0 license, please either i) remove the figure or ii) supply a replacement figure that complies with the CC BY 4.0 license. Please check copyright information on all replacement figures and update the figure caption with source information. If applicable, please specify in the figure caption text when a figure is similar but not identical to the original image and is therefore for illustrative purposes only.

Reviewers' comments:

Reviewer's Responses to Questions

**Comments to the Author**

1. Is the manuscript technically sound, and do the data support the conclusions?

Reviewer #1: Yes

Reviewer #2: Yes

2. Has the statistical analysis been performed appropriately and rigorously? 

Reviewer #1: Yes

Reviewer #2: I Don't Know

3. Have the authors made all data underlying the findings in their manuscript fully available?

Reviewer #1: No

Reviewer #2: No

4. Is the manuscript presented in an intelligible fashion and written in standard English?

Reviewer #1: Yes

Reviewer #2: Yes

5. Review Comments to the Author

Reviewer #1: Summary

This qualitative study explores health care provider’s experiences with managing people living with HIV, their perspectives on the barriers to adherence, and their recommendations. The objectives are clear and methodology is sound. The paper is easy to understand and follow. There are a few things that may need to be addressed to improve the paper as described below

General comments

The authors frequently use the terms defaulter and defaulted in the manuscript. These terms are not aligned with patient-centered language. Please consider revising these terms in the main text (excluding direct quotes from healthcare workers) to more patient-centered alternatives. For example, defaulter could be rephrased as person who has interrupted treatment, and defaulted as interrupted treatment.

Introduction

Line 64- Can the authors use the more updated UNAIDS 2024 HIV fact sheet for the global HIV statistics.

Figure 1- For consistency and clarity, can the authors consider changing caregiver’s perspectives to “healthcare provider” perspectives in the conceptual framework.

The aims of the study lack clarity regarding the study population. Was the focus on healthcare workers' perceptions of ART adherence limited to adult patients only, or did it include all populations (children, adolescents, and adults)?"

Although the literature on health care worker perspectives on ART adherence is scarce, it will be useful to review some existing literature to strengthen the study justification. You can consider adding these two articles:

Lahai M, Theobald S, Wurie HR, Lakoh S, Erah PO, Samai M, Raven J. Factors influencing adherence to antiretroviral therapy from the experience of people living with HIV and their healthcare providers in Sierra Leone: a qualitative study. BMC Health Serv Res. 2022 Nov 8;22(1):1327. doi: 10.1186/s12913-022-08606-x

Moucheraud C, Stern AF, Ahearn C, Ismail A, Nsubuga-Nyombi T, Ngonyani MM, Mvungi J, Ssensamba J. Barriers to HIV Treatment Adherence: A Qualitative Study of Discrepancies Between Perceptions of Patients and Health Providers in Tanzania and Uganda. AIDS Patient Care STDS. 2019 Sep;33(9):406-413. doi: 10.1089/apc.2019.0053. PMID: 31517526; PMCID: PMC6745526.

Methods

Typo in line 122- change “ast” to East.

Can the authors please specify whether the analysis was deductive or inductive.

Ethics- A more detailed discussion on how participant confidentiality and anonymity was maintained needs to be included in the ethics section.

Recruitment process- The recruitment process is unclear in terms of the number and type of healthcare workers that were targeted for the study. Furthermore, it is unclear why only 18 interviews were conducted. Was data collection stopped once saturation was reached?

Line 163- Please change the number 3 in bracket to 2.

Results

The authors have provided comprehensive data. To add depth to their narrative and strengthen the findings, it would be useful if they could also present any negative or outlier cases related to the identified challenges.

It would also be useful to provide a summary table that helps to better align the study recommendations to the reasons for interrupting ART. This could be done in a tabular format, where you have a column for the challenge, a second column describing the challenge, with a corresponding quote, and a third column with recommendations.

Discussion

The discussion is comprehensive and well-supported by evidence, but it includes some repetition as challenges and recommendations are addressed separately. Consolidating the findings by discussing challenges alongside healthcare provider recommendations in a single paragraph may better highlight the link between them.

Line 548, the sentence is incomplete.

Line 560- There is a repeated word “experiences”. Please delete the extra word.

Line 593- Please add the word “interruption” after ART.

The authors have listed some study limitations. Are there some strengths of the study that was conducted.

Conclusion

The conclusion is somewhat unclear. It would be beneficial if the authors could emphasize two or three main challenges and provide corresponding recommendations as a take-home message.

Reviewer #2: The manuscript provides a valuable contribution to the literature on antiretroviral therapy (ART) adherence by offering a unique perspective from healthcare providers, an often underrepresented yet critical stakeholder group. While much of the existing research focuses on patient-reported barriers and facilitators, this study highlights the systemic, logistical, and interpersonal challenges that healthcare providers observe in their interactions with patients. Focusing on the Volta Region of Ghana, it provides context-specific insights into the socio-cultural and operational barriers impacting adherence in resource-constrained settings. Additionally, the actionable recommendations offered such as decentralizing ART services, improving counselling, and addressing stigma are grounded in the practical realities of healthcare delivery, making the findings both academically significant and directly applicable to policy and practice. This approach bridges an important gap in the global effort to enhance ART adherence and achieve UNAIDS 95-95-95 targets.

The authors need to address the following:

Methods Section: Provide more detail on the sampling strategy, data saturation, and justification for the sample size.

• The manuscript mentions recruiting healthcare providers with ≥2 years of experience, which aligns with the study objectives. However, it lacks details about data saturation in interviews.

Limitations: Acknowledge methodological constraints like potential interviewer bias or regional contextual factors.

• The authors mention sample size and geographic scope as limitations. but a more comprehensive limitations section (e.g., interviewer bias, and generalizability of qualitative findings) would strengthen the manuscript.

Supplement with Appendices: Provide the interview guide and coding framework in appendices for transparency

6. PLOS authors have the option to publish the peer review history of their article (what does this mean? ). If published, this will include your full peer review and any attached files.

**Do you want your identity to be public for this peer review?** For information about this choice, including consent withdrawal, please see our Privacy Policy .

Reviewer #1: No

Reviewer #2: No

---

## [Author Response · Author response to Decision Letter 1]

27 Jun 2025

Response to Editor's comments

Response: The data availability statement has been improved as follows: “The raw data for this article will not be shared to protect the anonymity of the participants, as was agreed upon with the study participants. However, reasonable anonymous raw data can be requested from a third party within a period of five years via email: mananga@uhas.edu.gh, where the data will be kept on a dedicated desktop computer protected with a special password”.

Please ensure that you refer to Figure 2 in your text, as, if accepted, production will need this reference to link the reader to the figure.

Response: Figure 2 has been expunged from the work. Kindly refer to page 5 of the manuscript

---

## [Decision Letter · Decision Letter 1]

14 Sep 2025

PONE-D-24-41127R1Healthcare providers' experiences and recommendations for antiretroviral therapy adherence in Ghana: a facility-based phenomenological studyPLOS ONE

Dear Dr.Elvis Enowbeyang Tarkang:,

Thank you for submitting your manuscript to PLOS ONE. After careful consideration, we feel that it has merit but does not fully meet PLOS ONE’s publication criteria as it currently stands. Therefore, we invite you to submit a revised version of the manuscript that addresses the points raised during the review process.

We look forward to receiving your revised manuscript.

Kind regards,

Zewdu Gashu Dememew, M.D, PhD

Academic Editor

PLOS ONE

Journal Requirements:

Reviewers' comments:

Reviewer's Responses to Questions

**Comments to the Author**

1. If the authors have adequately addressed your comments raised in a previous round of review and you feel that this manuscript is now acceptable for publication, you may indicate that here to bypass the “Comments to the Author” section, enter your conflict of interest statement in the “Confidential to Editor” section, and submit your "Accept" recommendation.

Reviewer #1: All comments have been addressed

Reviewer #2: All comments have been addressed

2. Is the manuscript technically sound, and do the data support the conclusions?

Reviewer #1: Yes

Reviewer #2: Yes

3. Has the statistical analysis been performed appropriately and rigorously? 

Reviewer #1: N/A

Reviewer #2: Yes

4. Have the authors made all data underlying the findings in their manuscript fully available?

Reviewer #1: No

Reviewer #2: Yes

5. Is the manuscript presented in an intelligible fashion and written in standard English?

Reviewer #1: Yes

Reviewer #2: Yes

6. Review Comments to the Author

Reviewer #1: (No Response)

Reviewer #2: (No Response)

7. PLOS authors have the option to publish the peer review history of their article (what does this mean? ). If published, this will include your full peer review and any attached files.

**Do you want your identity to be public for this peer review?** For information about this choice, including consent withdrawal, please see our Privacy Policy .

Reviewer #1: No

Reviewer #2: No

---

## [Author Response · Author response to Decision Letter 2]

16 Sep 2025

Response to authors comments

Your study design and findings were well narrated and discussed with practical and feasible recommendations to HIV program in the country.

Response: Thank you for your comments.

You may consider a few of the following comments.

1.Line# 69: “ 95-95-95’’ Need to be revised as : ‘’At least 95% of people living with HIV know their HIV status, at least 95% of people who know their HIV status are on treatment, and at least 95% of people on treatment have a suppressed viral load”

Response: The sentence has been revised as suggested by the reviewer. Kindly refer to line #69-71.

2.Wasn’t it due to saturation reached at 18th participants that 18 health care were selected for the study (line #174)? Line # 188-189 seems to have another justification for 18 participants. Recheck these two sentences, please.

Response: This is a mishap. Sample size was based on data saturation as mentioned in line 174. the contradiction in line 188-189 has been deleted.

3.Check if secluded interview can be retained at line#187 or line # 200., to avoid repetition of same idea. Good to mention both but you may remain only under ethical issue.

Response: Lines 190-191 has been deleted from the work to avoid the repetition. It has only been retained under the ethical issue in line 200-201.

4.Please check if the statements under rigor might need references. E.g look at statements under lines#227-230.

Response: A reference has been provided for the section as suggested by the reviewer. Kindly refer to lines 220 and 228.

5.In lines#233-236 please try review and avoid repeated words such as ‘majority, most’. You may put only the %.

Response: Lines 234-237 has been revised. Repetition of words such has “majority” and “most” has been deleted.

6.Check the way you put the quotation mark under ‘Improvement in adherence to treatment’ and ‘stigmatization’.

Response: The sentence has been revise by removing the quotation marks in the middle of the quote. Kindly refer to Table 2. and line 301.

7.Line # 640: Good to suggest support from the government or NGOs. ART is to be taken for life where income generating activities (IGA) might be considered for sustained income.

Response. The paragraph has been revise to incorporate the idea suggested by the reviewer. Kindly refer to lines 648-649.

8.There have been a lot of PHIV perspective on ART adherence so far in Ghana (line 101), check if PLHIV perspective should be mentioned as limitation (line# 799).

Response: The mention of perspectives of PLHIV as a limitation has been expunged from the study as suggested by the reviewer. Kindly refer to lines 806-813.

---

## [Editor Report · Decision Letter 2]

18 Sep 2025

Healthcare providers' experiences and recommendations for antiretroviral therapy adherence in Ghana: a facility-based phenomenological study

PONE-D-24-41127R2

Dear Dr. Elvis Enowbeyang Tarkang  and the team,

We’re pleased to inform you that your manuscript has been judged scientifically suitable for publication and will be formally accepted for publication once it meets all outstanding technical requirements.

Kind regards,

Zewdu Gashu Dememew, M.D, PhD

Academic Editor

PLOS ONE
---

## [Editor Report · Acceptance letter]

PONE-D-24-41127R2

PLOS ONE

Dear Dr. Manu,

I'm pleased to inform you that your manuscript has been deemed suitable for publication in PLOS ONE. Congratulations! Your manuscript is now being handed over to our production team.

Kind regards,

on behalf of

Dr. Zewdu Gashu Dememew

Academic Editor

PLOS ONE